# Homeostasis after injury: How intertwined inference and control underpin post-injury pain and behaviour

Pranav Mahajan[1,2]*, Peter Dayan[2,3], Ben Seymour[1,4]

1 Nuffield Department of Clinical Neurosciences, University of Oxford, Oxford, United Kingdom, 2 Max Planck Institute for Biological Cybernetics, Tübingen, Germany, 3 University of Tübingen, Tübingen, Germany, 4 Institute of Biomedical Engineering, University of Oxford, Oxford, United Kingdom

* pranav.mahajan@ndcn.ox.ac.uk

## Abstract

Injuries are an unfortunate but inevitable fact of life, leading to an evolutionary mandate for powerful homeostatic processes of recovery and recuperation. The physiological responses of the body and the immune system must be coordinated with behaviour to allow protected time for this to happen, and to prevent further damage to the affected bodily parts. Reacting appropriately requires an internal control system that represents the nature and state of the injury and specifies and withholds actions accordingly. We bring the formal uncertainties embodied in this system into the framework of a partially observable Markov decision process. We discuss nociceptive phenomena in light of this analysis, noting particularly the counter-intuitive behaviours associated with injury investigation, and the propensity for transitions from normative, tonic, to pathological, chronic pain states. Importantly, these simulation results provide a quantitative account and enable us to sketch a much needed roadmap for future theoretical and experimental studies on injury, tonic pain, and the transition to chronic pain.

## Author summary

Several arguments suggest that the brain has a dedicated representation of the state of an injury. This provides an internal control system to modulate our behaviour, influencing the pain, anxiety, and mood changes appropriate to the need for heightened protection and recuperation during healing. Here, we propose a computational architecture for how this system might be constructed. We model the injury as a problem where its true state is only partially observable, requiring the brain to combine inference with optimal control to make decisions. We show how this framework offers an explanation of two core paradoxical observations: it explains behaviours such as rubbing or probing an injured area so as to gain information and reduce uncertainty, and it accounts for the high propensity of

**Data availability statement:** The code used in this paper is available at https://github.com/PranavMahajan25/InjuryPOMDP.

**Funding:** PM and BS have previously received a salary from Wellcome Trust (214251/Z/18/Z) and research funding from IITP (MSIT 2019-0-01371) and JSPS (22H04998). This research was also partly supported by the NIHR Oxford Health Biomedical Research Centre (NIHR203316). PM and BS are further hosted by the Wellcome Centre for Integrative Neuroimaging (WIN), which is funded by Wellcome Trust (203139/Z/16/Z and 203139/A/16/Z). PD is funded by and receives a salary from the Max Planck Society and the Humboldt Foundation. The funders had no role in study design, data collection and analysis, decision to publish, or preparation of the manuscript. The views expressed are those of the author(s) and not necessarily those of the NIHR or the Department of Health and Social Care.

**Competing interests:** The authors have declared that no competing interests exist.

an injury to transition into a pathological, chronic pain state via information restriction. Overall, this provides a quantitative framework for mapping the body's healing processes to their underlying neural substrates. This has the potential to help researchers identify novel targets for treating and possibly preventing chronic pain.

## Introduction

Injuries are common across the lifespan of many species and often lead to a period of vulnerability and reduced functionality whilst healing occurs. In both ecological and laboratory studies, characteristic behavioural changes such as increased anxiety and reduced activity are also often observed. They are considered to reflect adaptive changes and altered motivational priorities appropriate for safe recovery [1–3]. These changes are notoriously accompanied, and indeed potentially mediated, by forms of pain. Noting that organising appropriate behaviour is, by definition, a problem of control [4,5] offered a substantially new perspective on tonic pain. Here, we centre [5] in its natural reinforcement learning context, and provide the first concrete computational realisation of these ideas.

The critical innovation in [5] was to note that the brain suffers inevitable uncertainties about the nature and extent of the injury and the status of the recovery process. They suggested that the brain continually integrates multisensory and physiological inputs to infer, and thereby represent, the uncertain injury state. This representation would then be tied to the appropriate choice of action or inaction and generate internal signals interpreted as pain. This framework extends Bayesian models of pain perception [6–8] to address not only inferential aspects (for instance, expectancy effects) but also control under uncertainty. Crucially, they proposed that protective behaviours might restrict access to informative signals about recovery, potentially leading to persistent or maladaptive injury beliefs and contributing to chronic pain.

We formalise the suggestion of [5] as a partially observable Markov decision process (POMDP; [9]). This situates injury, recovery, and pain within the explanatory framework of neural reinforcement learning [4,10], where the solution is a policy mapping observations to actions or inaction. In doing so, our work addresses several gaps in the literature: it (i) unifies Bayesian inference and control approaches in pain research [11], (ii) mathematically formalises in a minimal model, how information restriction about whether injury has resolved, may drive chronic pain states [5], and (iii) makes explicit how the value of information influences pain-related behaviour [12].

We focus on the belief about the injury state constructed by the brain, proposing that this belief underlies the experience of tonic pain and modulates phasic pain responses. While this belief is informed by ascending sensory signals—particularly sustained nociceptor firing and hyperexcitability—such inputs, especially those from small-diameter, unmyelinated C-fibres [13], may be unreliable or imprecise, especially during the later stages of healing. Accurate injury state estimation therefore, requires integrating multiple sources of information, including autonomic and

physiological signals, exteroceptive cues such as vision [14], and prior expectations [5,11,15], accumulated over time to construct a coherent, control-relevant internal representation [7,8].

Although our simulations are inevitably limited, they show how tonic pain, as the belief about the state of the injury in the first instance, is a critical, internally constructed and interpreted signal which has broadcast properties, reorganising macroscopic patterns of behaviour. In our simulation results, we will dwell on a particular facet of the injury POMDP, which is the potentially large cost of the conventional policy solution to uncertainty, namely, information gathering to determine which options are better, or at least less worse. This cost could accrue if, for instance, the only way to learn about the current state of the injury is to attempt to use the injured body part. We examine such counter-intuitive actions of injury investigation (e.g. rubbing injured areas conventionally interpreted in the context of Gate Control Theory) in the light of our theory, and also suggest how the costs might lead to the development of pathological states of chronic pain, even when the periphery is apparently no longer presenting nociceptive input [16].

We proceed by building a simple POMDP model with just a handful of states and actions, and studying the properties of its optimal policy for different parameter values and starting states. The POMDP may seem rather abstract; we therefore tie it as closely as possible to phenomena observed in the context of pain.

## Results

### Theory sketch

We propose that the brain treats injury in terms of a partially observable Markov Decision Problem (POMDP; Fig 1A) [9, 17–19]. A POMDP comprises states $\mathcal{S}$, actions $\mathcal{A}$, transitions $\mathcal{T}$, observations $\mathcal{O}$ and utilities $\mathcal{R}$; we make substantial simplifications in our construction of all of these. The decision-theoretic task for the brain is then to determine a policy which maps the history of observations to the probability of taking an action to maximise the so-called future *return*, which is a long-run measure of cumulative utility. The general Bayesian Decision Theoretic framework is further described in the Methods section.

For the purpose of our simulations, we construct a simplified injury POMDP describing the following situation: A patient contemplating whether or not to commit to a demanding activity (e.g. getting groceries), whilst being uncertain about their injury. The true state characterises the current circumstance of the injury. We consider just two possible states $s_t \in \{0, 1\}$ for healthy or injured at time $t$. However, crucially, the brain lacks full information about the state – interoception is incomplete, as well as being noisy. The agent thus only has a so-called *belief* state $b_t \in [0, 1]$, which is a probabilistic distribution over all states [9]. The actions include everything that could be done; however, we make the radical simplification of considering just three to four actions: $a_{\text{act}}$, which involves anything physically demanding but can collect resources; $a_{\text{r\&r}}$, which allows time for recovery and recuperation; and $a_{\text{que}}$, which involves assessing the injury, for instance trying to walk on a recently broken leg (Fig 1B). We also allow for a null action $a_{\text{nul}}$ that does nothing (we omit this from a few experiments for simplicity). Transitions specify how states change, probabilistically, based on the actions – in general, with recovery and worsening of the injury, although here, we restrict ourselves to the case that the state actually remains constant (and so drop the time index for $s$). Observations include all the exteroceptive and interoceptive information available to the brain about the injury, requiring multisensory integration, but we reduce it to a single observation channel for simplicity in our simulations. Investigatory actions, such as probing the injured area, might provide both types of information.

Utilities are central to determining the optimal policy and are a contentious aspect of the POMDP. In standard RL, utilities are externally provided by the environment (e.g., points in a video game). However, in nature, animals must infer or construct affective consequences from observations alone. This has inspired work on homeostatic reinforcement learning [20] and intrinsic rewards [21–24]. Here, as a simplification, we assume the agent has an intrinsic reinforcement function $r(s, a)$, defining immediate affective consequences of action $a$ in true state $s$. For instance, this function might be large and negative/positive for activity/rest actions when injured (the former as a convenient proxy for the very long-run costs of incurring extra damage); small and negative for investigating while injured (due to potential harm); and negative for

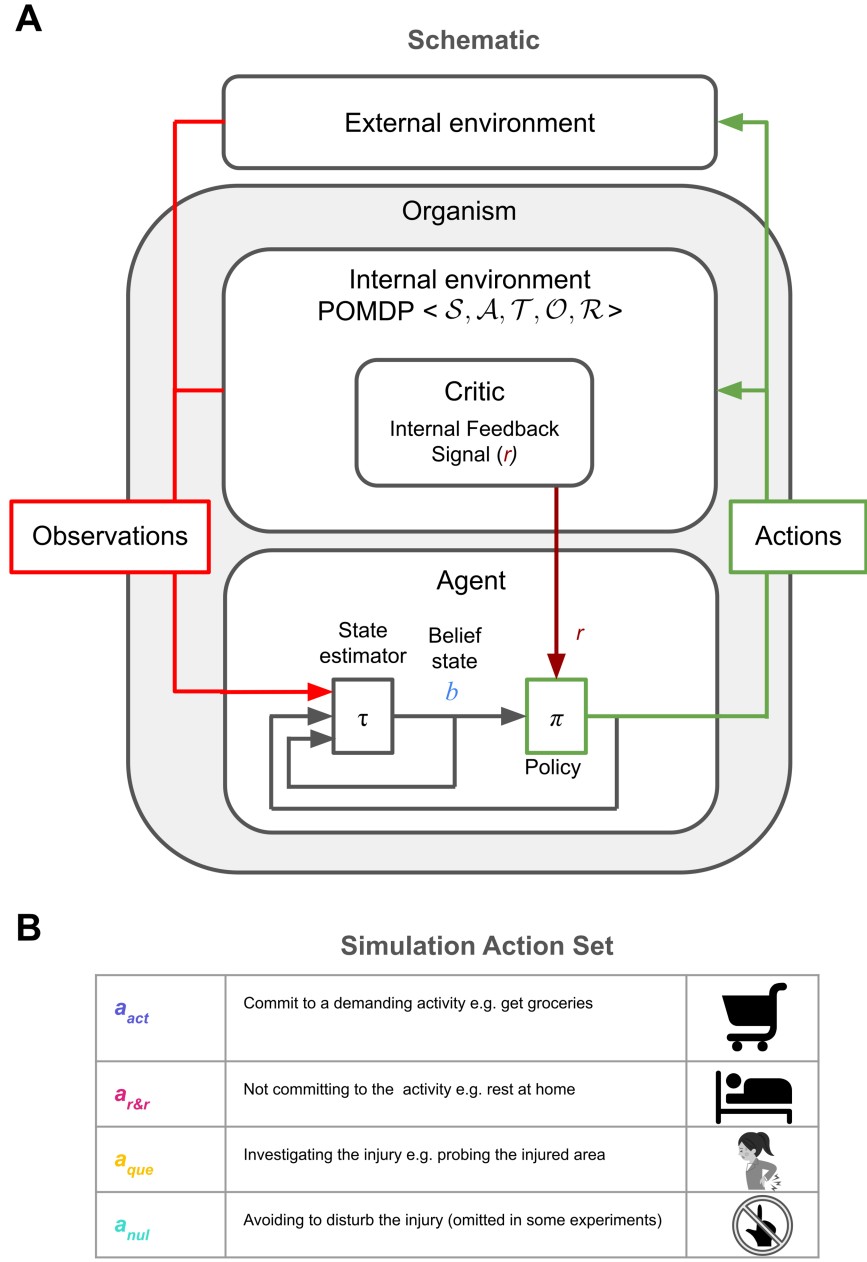

**A** Schematic

**B** Simulation Action Set

| | | |
|---|---|---|
| $a_{act}$ | Commit to a demanding activity e.g. get groceries | |
| $a_{r\&r}$ | Not committing to the activity e.g. rest at home | |
| $a_{que}$ | Investigating the injury e.g. probing the injured area | |
| $a_{nul}$ | Avoiding to disturb the injury (omitted in some experiments) | |

**Fig 1**. (A) Schematic of the injury POMDP, with an internal environment generating observations and conferring utilities and an internal agent inferring a belief state [9] to choose optimal actions. (B) The POMDP action set used for simulations.

resting while uninjured, reflecting unmet resource needs (again, as a proxy for the very long run effects of homeostatic deterioration). Since the agent only knows its belief $b_t$ about $s$, the expected utility is internally constructed.

We illustrate the injury POMDP in a simplified setting where the agent chooses a single, final, choice of $a_{act}$ or $a_{r\&r}$, but can choose to $a_{que}$ along the way if it is sufficiently uncertain about the injury state. The true state remains fixed throughout the decision-making episode, and the return is the cumulative utility of the trajectory of choices. The one-step expected utilities in the belief space of the injury POMDP used in our simulations are shown in Fig 2A. Note, that

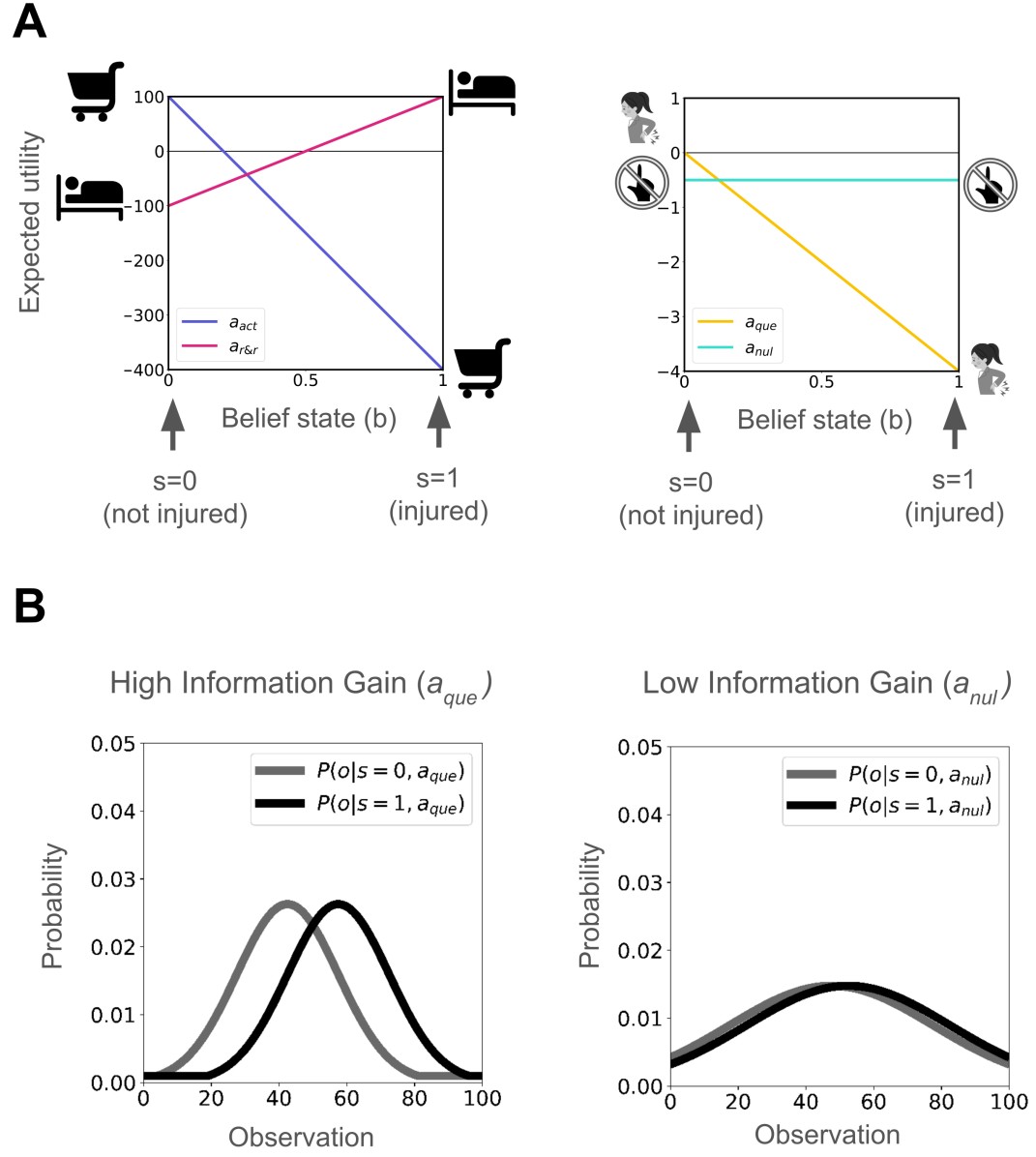

**Fig 2. Injury POMDP:** (A) Expected utilities of actions $a_{act}$, $a_{r\&r}$, $a_{que}$ and $a_{nul}$ in belief space. Utilities: when injured: $r(s=1, a_{act}) = -400$, $r(s = 1, a_{r\&r}) = +100$, $r(s = 1, a_{que}) = -4$; when uninjured: $r(s = 0, a_{act}) = +100$, $r(s = 0, a_{r\&r}) = -100$, $r(s = 0, a_{que}) = 0$. $r(a_{nul}) = -0.5$ regardless of the internal state as a minor opportunity cost. (B) Observation processes for actions $a_{que}$ and $a_{nul}$. (Further details in Methods - model simulation).

these expected utilities differ from long-run return for actions $a_{que}$ and $a_{nul}$. Choosing $a_{que}$ and $a_{nul}$ also results in sampling observations from distributions shown in Fig 2B. We see $a_{que}$ provides more discriminating or discerning observations about the internal state and is therefore more informative than $a_{nul}$. In our results, we often report state-action values $Q(b_t, a)$, which represent the expected long-run return for belief state $b_t$ and action $a$. The optimal action in any $b_t$ is the one with the highest $Q$ value.

For concreteness, as a substantial simplification, we associate the belief state $b_t$ with tonic pain – higher belief in being injured corresponds to greater pain. This pain becomes chronic if the agent fails to act or gather evidence to revise its

belief. We link the expected negative reinforcement from $a_{que}$ to phasic pain caused by injury investigation during the episode. This expectation averages over the belief state (see Methods), providing a mechanism for precisely tuning the feedback [12] in case of correct inferences, but is also susceptible to incorrect or underinformed inferences.

### Normative consequences

**Why do we investigate injury despite it being painful?**  Our model provides a normative explanation for the behaviour as to why we investigate our injury despite it being painful. This is difficult to explain through a simple inferential or a control-theoretic approach alone.

In this instance, the feedback for $a_{que}$ when injured is $r(s = 1, a_{que}) = -4$, denoting the negative consequence of probing the injury when injured. However, it also results in sampling observations using an observation process that can help infer the internal state more accurately. For this simulation demonstrating the value of information gained from injury investigation, we omit $a_{nul}$ for simplicity, but including it does not affect the results.

When starting from an uncertain belief state of $b_0 = 0.5$, the agent progressively samples observations by choosing $a_{que}$ at the cost of some (self-constructed) phasic pain. This is because there is a value to the information that can be gained given this uncertainty (see Fig 3). With every sample, the agent accumulates evidence about the internal state and updates its belief until it can commit to either action $a_{act}$ or $a_{r\&r}$ (Fig 3, red and blue arrows). The belief state acts as the context for driving optimal behaviour. When the true internal state is not injured $s = 0$, the agent updates its beliefs (red arrows) and chooses the optimal decision $a_{act}$ in the end. When the true internal state is injured $s = 1$, the agent updates its beliefs (blue arrows) and chooses the optimal decision $a_{r\&r}$ in the end. In conclusion, this demonstrates agent choosing injury investigating/probing actions despite associated with costs so as to accrue enough evidence to take the optimal decision. Note, actions investigating an injury extend beyond "rubbing the injured area" (explained by Gate Control Theory as "touch inhibits pain") to include non-contact explorations like bending a painful back or moving a painful joint etc.

**Trade-off between information gain and phasic pain:**  Consider the example of assessing your back after an injury. Mild discomfort from actions like stretching (e.g., "pins and needles") can provide valuable information about the injury's

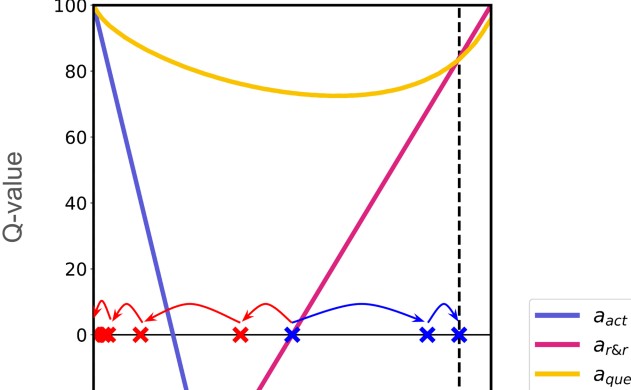

**Fig 3**. **Action values after value iteration.** Red arrows show belief updates under true state $s = 0$ (not injured), whereas blue arrows show belief updates under true state $s = 0$ (injured). The agent takes multiple $a_{que}$ actions, reaching either belief thresholds, where either choosing $a_{act}$ or $a_{r\&r}$ is more valuable than $a_{que}$ and thus terminating the episode.

state, outweighing the discomfort. In contrast, persistent tonic pain from bending offers little clarity about healing, as the sensory feedback may not distinguish between injury or recovery, making such actions less valuable.

We explored this trade-off between information gain and phasic pain using a 2×2 factorial simulation experiment. We manipulated the information gain by altering the observation process for $a_{que}$, using the two observation processes from Fig 2B. In this simulation, we omit $a_{nul}$ for simplicity of our didactic demonstration.

We find higher information gain of the observation process results in greater belief updates (Fig 4A), and thus confers a higher action value of $a_{que}$ across various uncertain belief states (Fig 4C). This contrasts with less informative observation process which results in small belief updates (Fig 4B) and lower value (Fig 4D). Further, when observation process is the same, higher cost of $a_{que}$ decreases the value of choosing $a_{que}$ (Fig 4C, 4D). This dynamic illustrates the trade-off between exploratory behaviour and pain avoidance. This experiment also highlights a key facet of our framework:

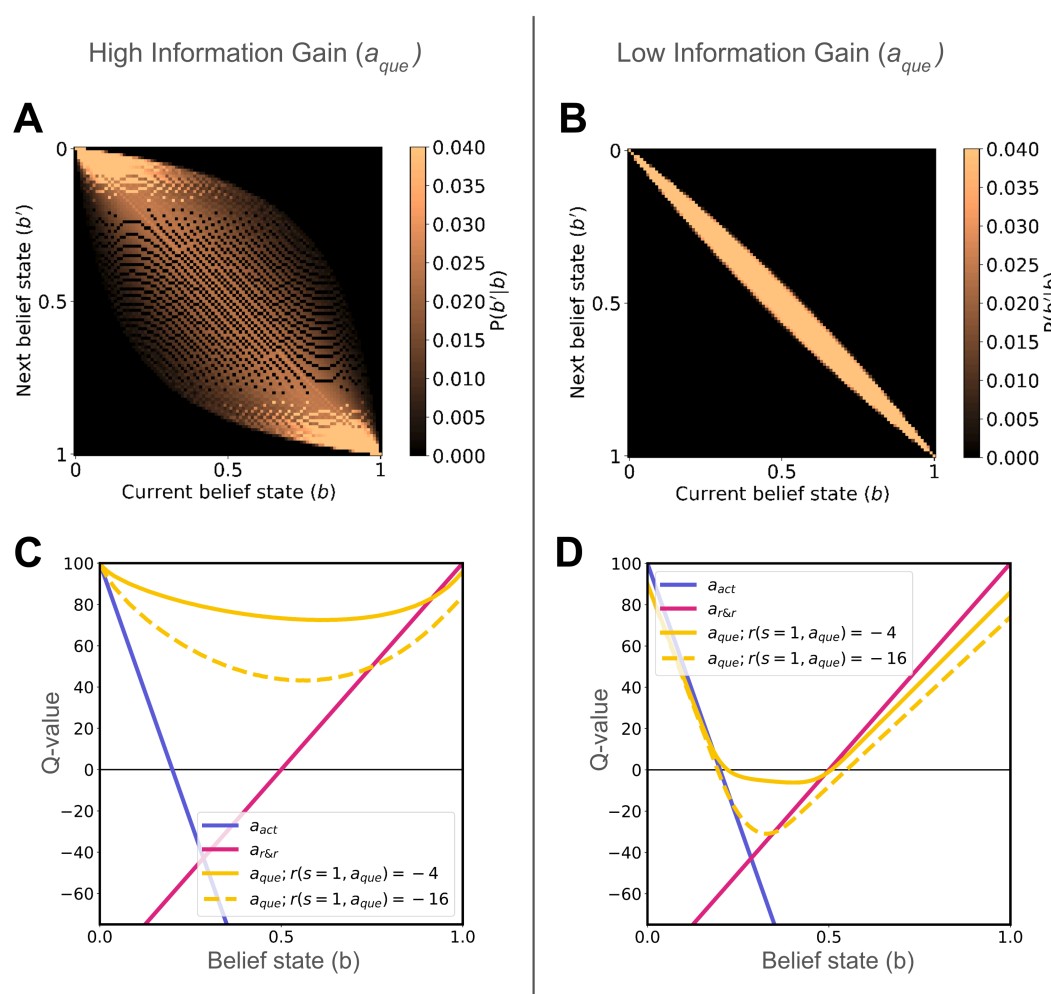

**Fig 4**. **Didactic demonstration of a 2×2 factorial design.** (A, B) Belief transitions from current belief state ($b$) to subsequent belief state ($b'$) reflect the influence of high vs. low informativeness of the observation process on belief updating. (C, D) Action value plots show how phasic pain ($r(s = 1, a_{que}) = -4$ vs. $r(s = 1, a_{que}) = -16$) and high vs low information gain influence $a_{que}$. Higher phasic pain reduces action value, while higher information gain increases value when the expected cost remains constant.

the informativeness of observations and the motivational (phasic) punishment used for credit assignment are disentangled/treated independently. In conclusion, the trade-off between information gain and phasic pain determines the value of an action in our model.

## Dysfunctional consequences

Having illustrated the normative consequences of the model, we will next consider how it might predict pathways to post-injury chronic pain. We describe two such pathways: (1) Information restriction and (2) Aberrant priors.

**Information restriction:** A person recovering from a back injury might reasonably avoid movement due to fear of pain. This would hinder them from gathering valuable information that could signal the resolution of the injury. Such a Fear-Avoidance [25] route to pain chronification has been elaborated in the path-dependent [26] suggestion of the information restriction hypothesis [5].

To investigate this route to pain chronification, we simulated the injury POMDP in which the true internal state is no longer injured ($s = 0$), as in the late stages of recovery. In this scenario, the action to investigate the injury ($a_{que}$) is nevertheless slightly painful when $b_t > 0$, due to the uncertainty of the injury. Action $a_{que}$ is more informative than $a_{nul}$, using the observation processes shown in Fig 2B. The action $a_{nul}$ has no cost associated with it, but is less informative than $a_{que}$. The difference from $a_{r\&r}$ is that it is not a terminating action - the agent could subsequently change its mind and execute one of the three other action.

We varied the cost of $a_{que}$ from $r(s = 1, a_{que}) = -4$ to $r(s = 1, a_{que}) = -16$, while keeping the observation processes fixed to explore different action values where information gain exceeds phasic pain and vice versa respectively (Fig 5A and 5B). We set the initial belief at the mid-point between the thresholds for choosing actions $a_{act}$ and $a_{r\&r}$. When the information gained from $a_{que}$ outweighs its associated pain, the individual's beliefs adaptively update, correctly resolving the hidden state to $b_t = 0$, leading ultimately to the optimal action ($a_{r\&r}$) over time (Fig 5C, averaged belief state trajectories). However, when the phasic pain of $a_{que}$ exceeds the information gain, it is less valued than $a_{nul}$ and $a_{nul}$ is preferred. Due to the less informative nature of $a_{nul}$, the beliefs do not change greatly. This leads to information restriction, as observed in belief state trajectories averaged over all simulation episodes (Fig 5D). Thus, we can observe a prolonged incorrect belief that the injury persists, leading to slower recovery or persistent injury mis-inference, which in our model correlates with increased tonic pain (as per Eq 11). Additionally, this behaviour results in more suboptimal avoidance behaviour - we see an increase in the agent choosing $a_{r\&r}$ (Fig 5D, right). This simulation illustrates how restricted information due to pain avoidance can contribute to the prolonged incorrect belief that the injury persists, providing a computational account for the chronic pain pathway described by [5].

**Aberrant priors:** We next study the adverse effects of having a strongly held incorrect belief about the severity of an injury, e.g. an aberrant prior. A prior (belief at the start of the decision-making episode) that underestimates the injury leads to further unnecessary harm, whereas a prior that overestimates the injury leads to over-protective behaviour. Both of our simulations use the same Q-values as Fig 5A.

In the first experiment, we use an underestimating prior when the true state is injured ($s = 1$). With aberrant starting belief states closer to zero (but greater than zero, as this is a decision threshold for $a_{act}$), we observe that the agent chooses $a_{que}$ over $a_{nul}$. This results in greater average cumulative expected costs (interpreted as cumulative phasic pain) accrued during the episode in the interest of accurate state inference and subsequent optimal decision-making (Fig 6A). These excess costs decrease as the starting belief becomes closer to $b_0 = 1$, i.e., being better aligned with the true state (Fig 6A). This demonstrates a pathway for excess phasic pain through injury investigation due to maladaptive priors underestimating an injury when the true state is injured.

In our second experiment, we use an overestimating prior when the true state is not injured ($s = 0$). If the starting belief state is aberrantly closer to 1 [itself a possible consequence of probability distortions in the context of risk aversion; [27]], then it contributes to prolonged incorrect belief that the injury persists, seen in belief trajectories in Fig 6B. Such prolonged

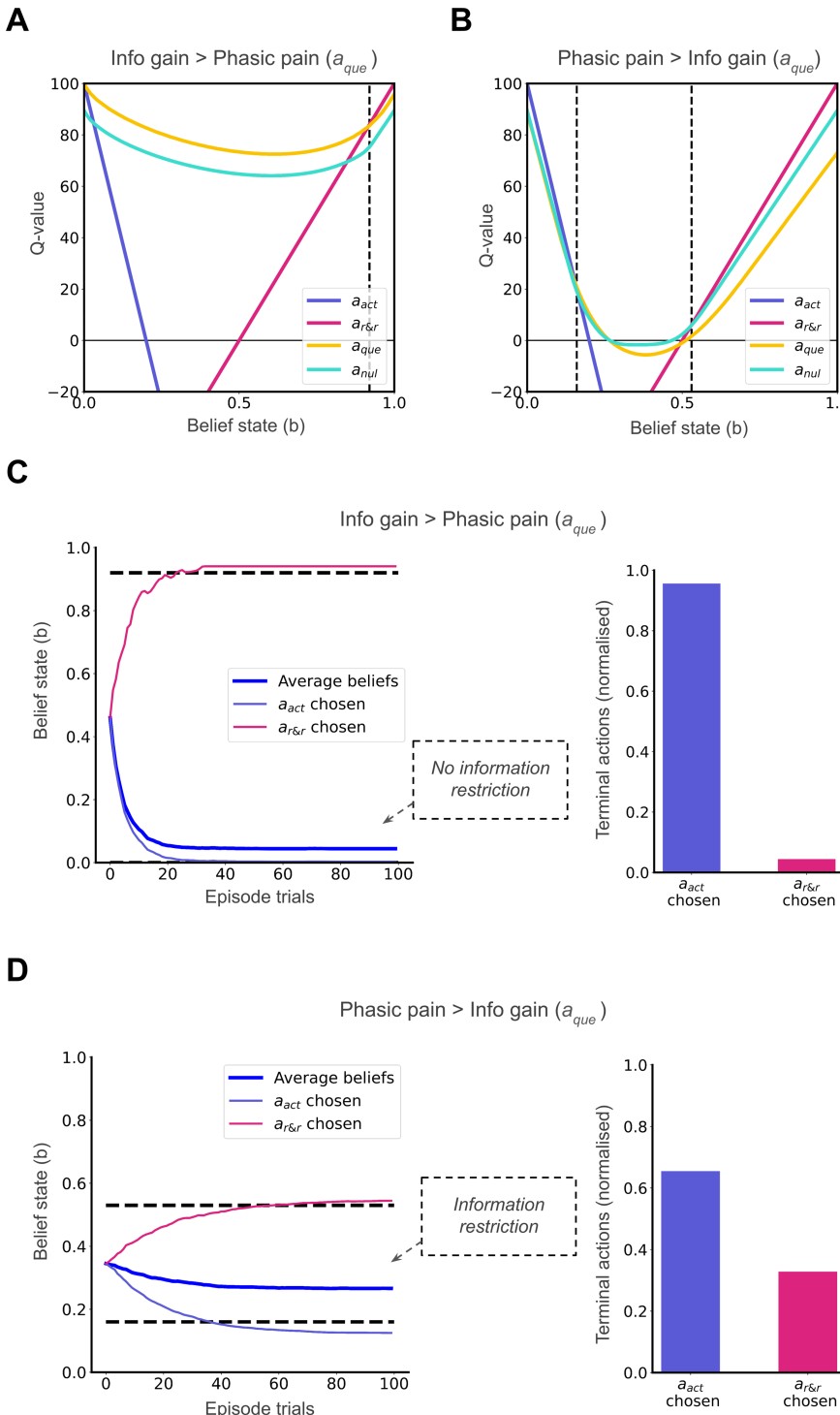

**Fig 5**. **Two possible scenarios along the recovery where the internal state is $s = 0$:** (A) Q-values when information gain from $a_{que}$ outweighs phasic pain costs; (B) Q-values when phasic pain costs outweigh information gain. (C) Faster adaptive hidden state resolution is seen when more informative $a_{que}$ is chosen over less informative $a_{nul}$ (D) Maladaptive information restriction occurs when less informative $a_{nul}$ is chosen over more informative $a_{que}$. Belief state trajectories are averaged over 1000 simulation episodes (dark blue) or only the episodes ending in $a_{act}$ (violet) and $a_{r\&r}$ (pink). 'Episode trials' represent steps within a decision-making episode.

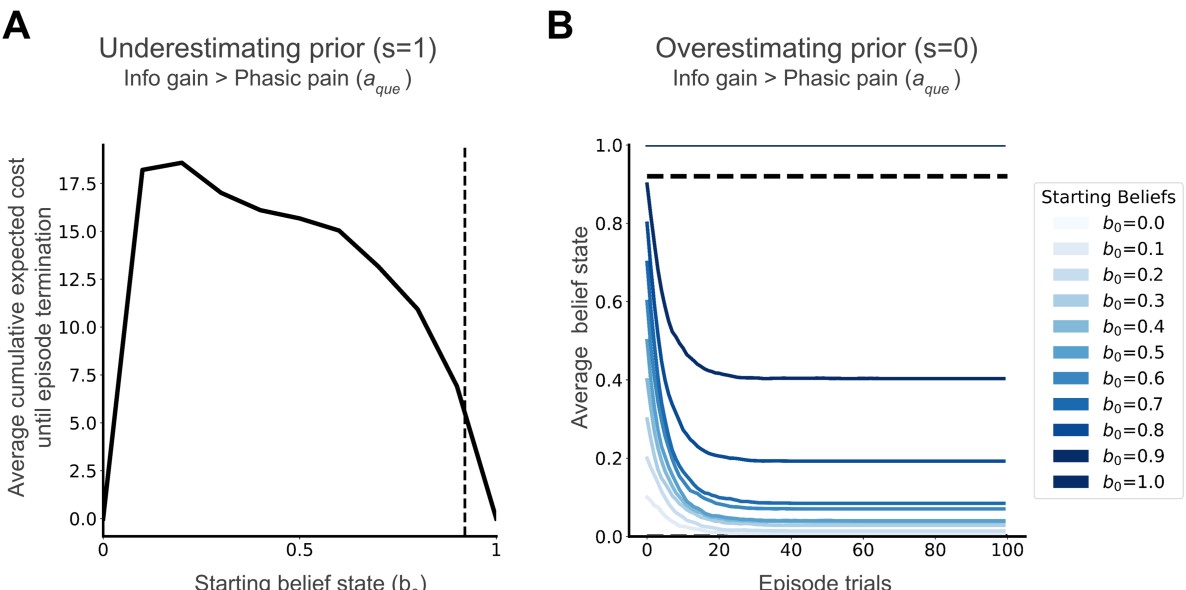

**Fig 6**. (A) Underestimation of injury when injured ($s = 1$), modelled as start belief states closer to 0, increases average phasic pain when injured. (B) Overestimation of injury when not injured ($s = 0$), modelled as start belief states closer to 1 but short of the decision threshold, can lead to prolonged incorrect beliefs, as seen in mean belief state trajectories. Belief state trajectories are averaged over 1000 simulation episodes.

incorrect belief decreases as the starting belief becomes closer to $b_0 = 0$, i.e., again being better aligned with the true state (Fig 6B). This demonstrates a pathway for persistent tonic and potentially chronic pain, due to maladaptive priors overestimating the injury when the true state is recovered.

## Discussion

We present a theoretical framework for understanding the computational logic of a dedicated homeostatic state for injury [5]. Central to this framework is the notion that internal states are partially observable and require inference. This perspective provides a normative explanation for behaviours such as probing an injury despite immediate phasic pain, as these actions prioritise information acquisition. Additionally, we identify two broad "fault lines" that can lead to suboptimal behaviour and chronic pain, offering insights into how the homeostatic injury system can go awry.

Our framework builds on prior models of homeostatic motivation [21–23] in which reinforcement signals are generated within the agent from observations, rather than being provided externally. However, we highlight the necessity of inferring internal states based on noisy observations and thus partial observability.

Our results align with aspects of the Fear-Avoidance model of pain chronification [28,29], offers a rich psychological framework grounded in fear learning and patient experience, positing that fear of movement leads to avoidance and physical deconditioning, thereby reducing opportunities to update beliefs about injury resolution. We offer the complementary approach (see Fig S1) of formalising information restriction and avoidance dynamics, providing a normative, computational, inference-based account that quantifies when and why avoidance could sustains maladaptive beliefs, consistent with theories of ex-consequentia reasoning [30–32]. Thus our model makes a precise, testable prediction about how avoidance behaviour (or its absence) can lead to persistent chronic pain (or recovery), respectively. This information-restriction pathway can be assessed in terms of stagnated belief updates about injury resolution. In contrast, the Fear-Avoidance model remains largely qualitative. While it clearly links avoidance to persistent fear, and lack of avoidance to fear extinction—drawing on insights from fear conditioning—its connection to chronic pain is more indirect. It relies on

the narrative that fear of pain or re-injury discourages activity, leading to physical deconditioning of muscles and disability and, ultimately, persistent pain [28]. Crucially, it thereby lacks such a formal account of how injury-related beliefs are inferred or updated.

Our simulations with aberrant priors, particularly those reflecting overestimation, also align with self-fulfilling prophecy and catastrophisation models [33,34], which highlight how catastrophic thinking enhances attentional demand and hinders disengagement from pain [35,36].

Looking ahead, the "fault lines" we identify can be categorised as the agent: solving the "wrong problem," solving the correct problem with the "wrong solution," or solving the correct problem correctly in an unfortunate, "wrong environment" [37]. For example, aberrant priors illustrate solving the wrong problem, while maladaptive behaviours, such as excessive avoidance due to Pavlovian biases [38] or habitual traits [39], represent wrong solutions that could lead to information restriction. A wrong environment may yield priors or utilities that were once adaptive but have become maladaptive, creating scenarios that appear to involve the wrong problem or solution. This categorisation provides a theoretical framework for linking current behavioural theories of chronic pain under a unified computational perspective, akin to computational nosology [40,41].

Our model is particularly relevant to clinical chronic pain conditions involving ambiguous symptoms, such as sciatica after lumbar radiculopathy. For instance, it may explain observations with sciatica muscle weakness patients, where baseline muscle weakness (a worse symptom) is counter-intuitively associated with improved outcomes [42,43]. Recovery of muscle strength could provide objective and unambiguous observations thereby providing high-information feedback for accurate injury inference and self-improvement. Similarly, our model could potentially explain phenomena such as boom-bust activity cycles [44,45] via faulty injury inference feeding into maladaptive behaviour. Potential interventions could involve improving injury inference, for example, through external cues or reducing maladaptive behaviour by activity pacing. Moreover, our framework may offer insights into the positive outcomes of Pain Reprocessing Therapy [46], particularly the role of somatic tracking and guided touch [47,48].

The model is just a first step. In particular, we did not include transitions in internal states, and thus actual recovery or exacerbation of injuries. Incorporating this could allow us to specify multiple 'activity' actions on a spectrum of intensity, with varying probabilities for worsening one's internal state, differential costs for acting more quickly, intensely or vigorously, and different times of completion. This will allow us to relate our model to notions of activity pacing, i.e. choosing appropriate intensity and timing, and also theories of vigour [49,50]. We rather arbitrarily generated the utility terms in our simulations; in the future, we plan to derive them from both physiological costs and the opportunity costs associated with the threats to homeostasis that forced inaction imposes. We have yet to model concrete neural or behavioural data (although we have identified ACL-tear as an ideal test case). More subtly, there are uncertainties about the exact modeling construct of tonic pain. Here, we modelled it directly using the belief over an injury state, which is in line with previous lines of work treating pain in terms of Bayesian inference. However, if an experiment is able to demonstrate that actions associated with information gain reduce tonic (as well as phasic) pain, as in endogenous analgesia [51,52] then it would deem value over belief states $V(b)$ as a more accurate correlate of tonic pain. Alternatively, one may also choose to represent tonic pain and its multiple attributes as a predictive state representation, as in general value functions [53]. Each of these various approaches implies a different meaning to tonic pain and its role in guiding behaviour.

The core contribution of this work is at the computational level, rather than an algorithmic one [54]. We use a model-based approach in this work, which is sufficient to account for the proposed phenomena. Model-free approaches, such as the actor-critic framework [55,56], may be able to reproduce the results, provided one uses recurrent neural networks (RNNs) as function approximators. These can learn to represent belief information correctly as long as the RNN capacity is sufficient [57]. However, a large amount of interaction data under different scenarios is typically required to train these RNNs and allow them to operationalise the value of information gain, e.g. meta-learning [58,59]. This data requirement presents a challenge, as taking precarious actions (e.g., repeatedly acting when severely injured) risks worsening the injury or, in the worst case, causing death, which would preclude learning altogether. Therefore, it is likely that we are

endowed with innate models that are either fine-tuned through experience or used to guide model-free learning. Note further that in the case of actor-critic models, only the critic units would be expected to represent the value of information gain.

Finally, and perhaps of greatest translational importance, chronic pain likely arises from complex interactions between learning processes, internal and external environments, and behavioural feedback loops. By offering a computational framework that spans from phasic to chronic pain, our account provides a mathematical foundation for making theories of this condition explicit.

## Methods

### Bayesian decision theoretic approach to homeostasis

We build upon the Bayesian Decision Theoretic (BDT) framework [10,37] utilising POMDPs [9,17–19,60] and extend it to the problem of homeostasis.

Our internal environment POMDP is formally represented as a tuple $\langle \mathcal{S}, \mathcal{A}, \mathcal{T}, \mathcal{O}, \mathcal{R} \rangle$, where: $\mathcal{S}$ is the set of all internal states ($s$). $\mathcal{A}$ is the set of all possible actions ($a$). $\mathcal{T}$ is the transition function, where $\mathcal{T}(s, a, s') = P(s'|s, a)$ is the probability of ending in $s'$ by performing action $a$ in state $s$. $\mathcal{O}$ is the set of all observations ($o$), where the observation function is $\mathcal{O}(s', a, o) = P(o|a, s')$ is the probability of observing $o$ if action $a$ is performed and resulting state is $s'$. $\mathcal{R}$ are the utilities, where $r(s, a)$ is the internal feedback signal obtained by taking action $a$ in internal state $s$.

At any time $t$, the agent does not necessarily have access to its complete internal state $s_t$, but has access to observations $o_t$ from the internal environment, which provide incomplete (noisy) information about the internal state. The agent therefore infers a belief state $b_t$ over internal state space $\mathcal{S}$ [18]. The belief state is defined as the posterior probability density of being in each internal state $s$, given the complete history $h_t$ of actions and observations at any time $t$, $h_t = \{a_0, o_1, ...o_{t-1}, a_{t-1}, o_t\}$ and initial belief $b_0$

$$b_t(s) = P(s_t = s|h_t, b_0) \tag{1}$$

The belief state $b_t$ is a sufficient statistic for the history $h_t$ [61,62]. At any time $t$, the belief state $b_t$ can be computed from the previous belief state $b_{t-1}$, using the previous action $a_{t-1}$ and the current observation $o_t$ using a state estimator function ($\tau$) derived using Bayes rule [9,60]

$$b_t(s') = \tau(b_{t-1}, a_{t-1}, o_t)(s')$$
$$= \frac{\mathcal{O}(s', a_{t-1}, o_t) \sum_{s \in \mathcal{S}} \mathcal{T}(s, a_{t-1}, s') b_{t-1}(s)}{P(o_t|b_{t-1}, a_{t-1})} \tag{2}$$

where the $P(o_t|b_{t-1}, a_{t-1})$ is the marginal distribution over observations $o_t$ (independent of $s'$, acting as a normalising factor for $b_t$). This belief transition function ($\tau$) is used to obtain belief transition plots and belief trajectories.

A critical concept for the POMDP is a stationary policy $\pi(b, a)$ which is a probability distribution over the action $a$ that the agent will take given that its current belief is $b$. Define $V^\pi(b_t)$ as the expected discounted long-term reward following policy $\pi$ as the value of this belief state:

$$V^\pi(b) = \mathbb{E}\left[\sum_{t=0}^{\infty} \gamma^t \sum_{a_t \in \mathcal{A}} \rho(b_t, a_t)\pi(b_t, a_t)|b_0 = b\right] \tag{3}$$

where $\gamma$ is a temporal discount function and

$$\rho(b_t, a_t) = \sum_{s \in \mathcal{S}} b_t(s) r(s, a_t) \qquad (4)$$

is the feedback function on belief states, constructed from the original feedback signal on internal states $r(s, a)$. Then, it can be shown that the value satisfies the Bellman equation:

$$V^{\pi}(b) = \sum_{a \in \mathcal{A}} \pi(b, a) \left[ \rho(b, a) + \gamma \sum_{o \in \mathcal{O}} P(o|b, a) V^{\pi}(\tau(b, a, o)) \right] \qquad (5)$$

where the sum over $\mathcal{O}$ is interpreted as the expected future return over the infinite horizon of executing action $a$, assuming the policy $\pi$ is followed afterwards. It is known that there is a deterministic policy $\pi^*(b, a)$ that optimizes the value of all belief states [60]

$$\pi^*(b) = \arg \max_{\pi \in \Pi} \mathbb{E} \left[ \sum_{t=0}^{\infty} \gamma^t \sum_{a \in \mathcal{A}} \rho(b_t, a) \pi(b_t, a) | b_0 = b \right] \qquad (6)$$

The value function $V^*$ of the optimal policy $\pi^*$ is the fixed point of the Bellman's equation [63]

$$V^*(b) = \max_{a \in \mathcal{A}} \left[ \rho(b, a) + \gamma \sum_{o \in \mathcal{O}} P(o|b, a) V^*(\tau(b, a, o)) \right] \qquad (7)$$

and one can define a corresponding optimal Q-value function, which refers to the value of taking action $a$ and then following the optimal policy, as

$$Q^*(b, a) = \rho(b, a) + \gamma \sum_{o \in \mathcal{O}} P(o|b, a) V^*(\tau(b, a, o)) \qquad (8)$$

## Model implementation and simulation details

We do not aim to model the entire lifetime of an agent but rather an episode in the agent's lifetime. Here, we maintain a binary hidden state $s = \{0, 1\}$ and we do not model transitions between states. Since we do not model transitions in $s_t$ within the POMDP episode, the state estimator Eq 2 defining the belief transitions is reduced to the following,

$$b_t(s) = \tau(b_{t-1}, a_{t-1}, o_t)(s) = \frac{\mathcal{O}(s, a_{t-1}, o_t) b_{t-1}(s)}{P(o_t | b_{t-1}, a_{t-1})} \qquad (9)$$

This is similar to the Tiger POMDP environment by [9], except that the $a_{\text{que}}$ is also belief-dependent. Further we refer to the belief state over $s$ as $P(s = 1) = b_t(s = 1) = b$ and $P(s = 0) = b_t(s = 0) = 1 - b$. If the POMDP has more states, the belief state provides the probability distribution of being in each of those states.

To solve the POMDP, we treat it as a belief MDP, and of the several approaches to perform value approximation [64, 65], we utilise belief grid value iteration [66] leading to piecewise linear and convex (PWLC) value functions. Here, we discretise the belief space in steps of 0.01 and the observation distribution, $o \in (0, 100] = \mathcal{O}$, in steps of 1 and assume the observation function to be known.

From the action set $\mathcal{A} = \{a_{act}, a_{r\&r}, a_{que}, a_{nul}\}$ described in Fig 1, $a_{act}$ and $a_{r\&r}$ result in episode termination and $a_{que}$ and $a_{nul}$ do not. We then used the Q values of each action in each belief state to determine the decision thresholds for choosing actions $a_{act}$ and $a_{r\&r}$ which terminate the episode, defined as the most uncertain belief state $b$ where the optimal action is no longer a non-terminal action.

**Simulation parameters**

| Experiment | True state | Outcome for $a_{que}$ when injured | Observation model for $a_{que}$ |
|---|---|---|---|
| Injury investigation | $s = 1$ | $r(s = 1, a_{que}) = -4$ | $O(s, a_{que}, o) = HI$ |
| Phasic pain - information gain trade-off (High information gain condition) | $s = 1$ | $r(s = 1, a_{que}) = -4, -16$ | $O(s, a_{que}, o) = HI$ |
| Phasic pain - information gain trade-off (Low information gain condition) | $s = 1$ | $r(s = 1, a_{que}) = -4, -16$ | $O(s, a_{que}, o) = LI$ |
| Information restriction (Info gain >phasic pain) | $s = 0$ | $r(s = 1, a_{que}) = -4$ | $O(s, a_{que}, o) = HI, O(s, a_{nul}, o) = LI$ |
| Information restriction (Phasic pain >info gain) | $s = 0$ | $r(s = 1, a_{que}) = -16$ | $O(s, a_{que}, o) = HI, O(s, a_{nul}, o) = LI$ |
| Underestimating prior | $s = 1$ | $r(s = 1, a_{que}) = -4$ | $O(s, a_{que}, o) = HI, O(s, a_{nul}, o) = LI$ |
| Overestimating prior | $s = 0$ | $r(s = 1, a_{que}) = -4$ | $O(s, a_{que}, o) = HI, O(s, a_{nul}, o) = LI$ |

The utilities mentioned in Fig 2 were fixed throughout, with the exception of experiments where phasic pain exceeds information gain for $a_{que}$, therefore, we vary $r(s = 1, a_{que}) = -4$ to $r(s = 1, a_{que}) = -16$. The observation process for $a_{que}$ and $a_{nul}$ were always high information gain (HI) and low information gain (LI) respectively, as shown in Fig 2, with the exception of the Fig 4, where we vary the information gain of $a_{que}$. For the HI process, the means of observation distributions for $s = 0$ and $s = 1$, were apart by 15 units and the observation distributions had the std. dev. of 15 units. For the LI process, the means of observation distributions for $s = 0$ and $s = 1$, were apart by 5 units and the observation distributions had the std. dev. of 30 units. Action $a_{nul}$ was omitted from Normative Consequences results for simplicity but included in Dysfunctional Consequences results. The maximum trials within a POMDP episode were set to 100. Results were averaged over 1000 independent POMDP episodes for a fixed utility function. The discount factor was set to $\gamma = 1$.

**Definition of phasic and tonic pain in our model simulations**

In the first instance, we define phasic pain based on Eq 4, as belief-dependent negative feedback (expected negative costs of an action), only when concerning the pain/injury systems.

$$\text{Phasic Pain} \propto \sum_{s \in \mathcal{S}} b(s)\zeta(s, a) \tag{10}$$

Here, $\zeta(s, a)$ is the negative rewards or punishments concerning the pain systems and excludes other kinds of punishments such as opportunity costs or loss in resources. In our simulations, we only require and report the phasic pain for action $a_{que}$, where the $\zeta(s, a_{que}) = r(s, a_{que})$.

We define tonic pain to be proportional to the belief-weight drive in the injury state space. This further contributes to the affective value, which is a function of the belief state $V^\pi(b)$.

$$\text{Tonic Pain} \propto b(s = 1) \tag{11}$$

## Acknowledgments

PM would like to thank Michael Browning, Rafal Bogacz, Suyi Zhang, Charlie Yan, Shuangyi Tong, and Danielle Hewitt for their feedback on presentations or earlier drafts of the manuscript. For the purpose of open access, the authors have applied a CC BY public copyright licence to any Author Accepted Manuscript version arising from this submission.

## Supporting information

**S1 Fig. Comparison of predictions of our homeostatic injury state model to that of the Fear Avoidance model and how they play a complementary role in educating patients about their pain.**
(TIFF)

## Author contributions

**Conceptualization:** Pranav Mahajan, Peter Dayan, Ben Seymour.

**Funding acquisition:** Ben Seymour.

**Investigation:** Pranav Mahajan.

**Methodology:** Pranav Mahajan.

**Software:** Pranav Mahajan.

**Supervision:** Peter Dayan, Ben Seymour.

**Visualization:** Pranav Mahajan.

**Writing – original draft:** Pranav Mahajan.

**Writing – review & editing:** Pranav Mahajan, Peter Dayan, Ben Seymour.

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
