## [Decision Letter · Decision Letter 0]

1 Aug 2025

PCOMPBIOL-D-25-01015

Homeostasis After Injury: How Intertwined Inference and Control Underpin Post-Injury Pain and Behaviour

PLOS Computational Biology

Dear Dr. Mahajan,

Thank you for submitting your manuscript to PLOS Computational Biology. After careful consideration, we feel that it has merit but does not fully meet PLOS Computational Biology's publication criteria as it currently stands. Therefore, we invite you to submit a revised version of the manuscript that addresses the points raised during the review process.

Please submit your revised manuscript within 30 days Oct 01 2025 11:59PM. If you will need more time than this to complete your revisions, please reply to this message or contact the journal office at ploscompbiol@plos.org. Please include the following items when submitting your revised manuscript:

We look forward to receiving your revised manuscript.

Kind regards,

Christoph Mathys

Academic Editor

PLOS Computational Biology

Hugues Berry

Section Editor

PLOS Computational Biology

**Journal Requirements:**

At this stage, the following Authors/Authors require contributions: Pranav Mahajan, Peter Dayan, and Ben Seymour. Please ensure that the full contributions of each author are acknowledged in the "Add/Edit/Remove Authors" section of our submission form.

5) Please ensure that the funders and grant numbers match between the Financial Disclosure field and the Funding Information tab in your submission form. Note that the funders must be provided in the same order in both places as well.

**Reviewers' comments:**

Reviewer's Responses to Questions

**Comments to the Authors:**

Reviewer #1: This paper extends Seymour et al. (2023) by implementing a formal POMDP-based model in which the brain infers injury states to guide behaviour. The authors use simulations to show that rational behaviours emerge when observations are informative, while maladaptive outcomes (such as chronic pain) arise from information restriction or aberrant priors. This generative modelling framework offers a computational approach to understanding injury-related pain and recovery.

The paper presents two central findings. First, it shows that even when investigating an injury is painful, such behaviour can be rational and beneficial. Within the proposed computational model, this kind of action helps reduce uncertainty about the injury state, allowing the brain to make better long-term decisions about whether to rest or resume activity. Second, the authors demonstrate how chronic pain can emerge through two distinct pathways: either by avoiding information-gathering actions due to fear of pain, leading to persistent uncertainty, or by starting with incorrect prior beliefs about the severity of the injury. Both scenarios can result in prolonged maladaptive behaviour, offering a formal explanation for the transition from acute to chronic pain in the absence of ongoing tissue damage. These findings are intuitive, and they resonate with common sense understandings of how we behave when hurt.

I have previously reviewed this paper as a CCN conference paper, and my earlier comments have been addressed in the present version. My only additional comment is a suggestion to add a more informative README file in the GitHub repository, and to ensure that all code is clearly commented, as this is not the case at the moment.

Reviewer #2: The article discusses a formalisation of the state of injury in terms of a POMDP; which allows to gain insights into the behavioural processes to aid with recovery, and into how these are shaped by uncertainty about the underlying latent injury state. The paper is well written and well structured. It also acknowledges it being a first step in a somewhat complex and multifaceted modelling endeavour. I think this is an excellent contribution and would appreciate this being published on this outlet. I only have two conceptual notes.

(1) One remark concerns the apparent elementary nature of the model, which penalises its potential for prediction outside of the running example. Specifically, acting usually falls along a spectrum of intensity: take for example the issue of someone who professionally trains e.g. grip strength and needs to both recover from an injury, but also keep their strength from degrading (because it is functional to their career). Their choice becomes one about training intensity (e.g. when to train, and how much training to do). Could the authors perhaps discuss how the model would extend further beyond punctate actions and deal with a dimensional action range? and perhaps trace a connection to the computational theory of vigour ?

(2) Another conceptual note is that while the approach presented is definitely sufficient to account for known phenomena, which of its components are necessary and critical remains slightly unclear. For instance it would be lovely to see the present framework contrasted with a fully model-free approach to learning about the state of injury. One possibility would just be to use a bare actor critic framework, much like Maia's actor critic to explain avoidance conditioning (Maia, T. V. (2010). Two-factor theory, the actor-critic model, and conditioned avoidance. Learning & behavior, 38(1), 50-67.) just revamped for this specific context. What would be the phenomena that are not explained (or explained poorly) in a fully model-free context that are instead well captured here?

**Have the authors made all data and (if applicable) computational code underlying the findings in their manuscript fully available?**

Reviewer #1: Yes

Reviewer #2: None

PLOS authors have the option to publish the peer review history of their article (what does this mean?). If published, this will include your full peer review and any attached files.

Reviewer #1: No

Reviewer #2: No

**Figure resubmission:**
---

## [Decision Letter · Decision Letter 1]

19 Sep 2025

Dear Mr. Mahajan,

We are pleased to inform you that your manuscript 'Homeostasis After Injury: How Intertwined Inference and Control Underpin Post-Injury Pain and Behaviour' has been provisionally accepted for publication in PLOS Computational Biology.

Best regards,

Christoph Mathys

Academic Editor

PLOS Computational Biology

Hugues Berry

Section Editor

PLOS Computational Biology

Reviewer #1:

Reviewer #2:

Reviewer's Responses to Questions

**Comments to the Authors:**

Reviewer #1: I appreciate the effort the authors have put into improving the repository. The revisions effectively address the concerns I had previously raised, and I am happy with the revised manuscript.

Reviewer #2: The authors have addressed my comments satisfactorily.

**Have the authors made all data and (if applicable) computational code underlying the findings in their manuscript fully available?**

Reviewer #1: None

Reviewer #2: None

PLOS authors have the option to publish the peer review history of their article (what does this mean?). If published, this will include your full peer review and any attached files.

Reviewer #1: No

Reviewer #2: No

---

## [Editor Report · Acceptance letter]

PCOMPBIOL-D-25-01015R1

Homeostasis After Injury: How Intertwined Inference and Control Underpin Post-Injury Pain and Behaviour

Dear Dr Mahajan,

I am pleased to inform you that your manuscript has been formally accepted for publication in PLOS Computational Biology. Your manuscript is now with our production department and you will be notified of the publication date in due course.

With kind regards,

Zsofia Freund
